# PersonaLM: Language Model Personalization via Domain-distributed Span Aggregated K-Nearest N-gram Retrieval Augmentation

**Puneet Mathur**[‡][*], **Zhe Liu**[†]**, Ke Li**[†]**, Yingyi Ma**[†]**, Gil Keren**[†]**, Zeeshan Ahmed**[†],
**Dinesh Manocha**[‡]**, Xuedong Zhang**[†]

‡ University of Maryland, College Park

† Meta

‡{puneetm,dmanocha}@umd.edu

†{zheliu,kli26,yingyima,gilkeren,ahzee,xuedong}@meta.com

## Abstract

We introduce **PersonaLM** - Domain-distributed span-Aggregated K-nearest N-gram retrieval augmentation to improve language modeling for Automatic Speech Recognition (ASR) personalization. PersonaLM leverages contextually similar n-gram word frequencies for recognizing rare word patterns associated with unseen domains. It aggregates the next-word probability distribution based on the relative importance of different domains to the input query. To achieve this, we propose Span Aggregated Group-Contrastive Neural (SCAN) retriever that learns to rank external domains/users by utilizing a group-wise contrastive span loss that pulls together span representations belonging to the same group while pushing away spans from unrelated groups in the semantic space. We propose ASAP benchmark for ASR LM personalization that consists of three user-specific speech-to-text tasks for meetings, TED talks, and financial earnings calls. Extensive experiments show that PersonaLM significantly outperforms strong baselines with a 10-16% improvement in perplexity and a 5-8% reduction in Word Error Rates on popular Wikitext-103, UserLibri, and our ASAP dataset. We further demonstrate the usefulness of the SCAN retriever for improving user-personalized text generation and classification by retrieving relevant context for zero-shot prompting and few-shot fine-tuning of LLMs by 7-12% on the LAMP benchmark.

## 1 Introduction

Language modeling is a core task in NLP with important applications in automatic speech recognition (ASR) (Mikolov et al., 2010; Chen et al., 2015; Xu et al., 2018). Pre-trained LMs (Irie et al., 2019a; Li et al., 2020a) memorize a surprising amount of knowledge from their training corpora in the underlying neural network parameters(Petroni et al., 2019; Jang et al., 2022). However, this makes it difficult to personalize them for text generation, non-

---

[*]Work done during internship at Meta

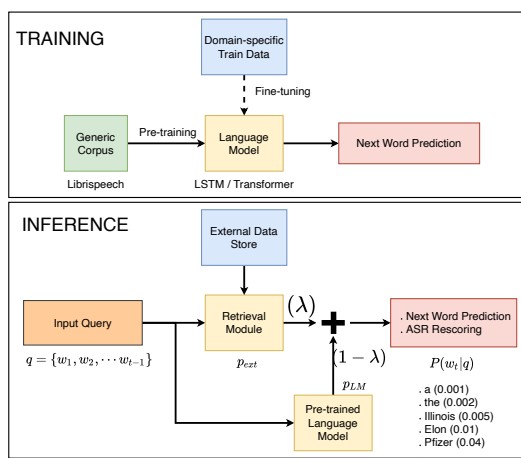

Figure 1: ASR Personalization: During training, the LM is pre-trained on a generic corpus and optionally fine-tuned on the out-of-domain corpus (see dotted). For query $q$ at inference, LM output $p_{LM}$ is interpolated with the probability distribution $p_{ext}$ retrieved from domain-specific external corpus for next word prediction $p(w_t|q)$ and ASR re-scoring.

streaming ASR re-scoring, and on-device streaming ASR models for unseen users and domains due to the existence of user-preferred rare word patterns, facts, proper names, and other domain-specific tail words not seen frequently in the training data (Schick and Schütze, 2019; Maynez et al., 2020; Serai et al., 2022). Retrieval augmentation (Lewis et al., 2020) (see Fig. 1) can help personalize LMs by explicitly exposing them to external world knowledge during inference (Borgeaud et al., 2022). LMs leverage the retrieval mechanism to select contextually relevant users/domains from an external corpus and then attend over that knowledge to inform their predictions (Liu et al., 2022).

Prior research has explored kNN-LM memorization (Khandelwal et al., 2020), RETRO (Borgeaud et al., 2022), and attention-based caches (Grave et al., 2017). However, these methods are not suited for retrieving relevant domains/users prior to context selection, rather were proposed primarily to enhance the memorization capabilities of LM. Recent approaches, notably REALM (Guu et al., 2020)

and RAG (Lewis et al., 2020), incorporate a non-parametric retrieval step during LM pre-training, thus being unable to adapt their context representation for unseen domains.

We address the challenge of capturing rare word patterns associated with specific users/domains by exploiting n-gram word frequencies from underlying domains. Further, we hypothesize that n-gram patterns are domain/user-specific, and augmenting LM predictions with n-gram probabilities from a subset of query-relevant users/domains may lead to better personalization. We anticipate the retrieval augmentation through n-gram frequencies to have additional advantages of very low computational overhead, efficient caching, and asynchronous updates for newer data without the need for re-computation from scratch.

We propose PersonaLM – Domain-distributed Span-aggregated k-Nearest N-gram Language Model that aggregates top-k nearest n-gram co-occurrence frequencies from each domain weighted according to the domain's relative importance to the input query, which is augmented with the target word probability distribution for next word prediction and ASR second-pass re-scoring. We utilize a novel Span Aggregated Group-Contrastive Neural (SCAN) retriever that can learn highly discriminative semantic representations to distinguish between text spans from the same group as opposed to random spans using a group-wise contrastive loss. SCAN retriever assigns a relevance score to each textual document/recording from an external corpus based on its semantic similarity with the input query to weigh their contribution to the final prediction. **Our main contributions are**:

- **PersonaLM** retrieval augmentation for ASR personalization that leverages group-wise contrastive loss to train **Span Aggregated Group-Contrastive Neural (SCAN) retriever** for ranking query-relevant external domains/users and augments domain-distributed k-nearest n-gram frequencies to improve LM predictions.
- **ASAP - a novel benchmark for ASR LM personalization** consisting of three user-specific ASR tasks in the domains of meetings, TED talks, and financial conference calls. PersonaLM significantly outperforms strong baselines on ASAP benchmark, UserLibri, and Wikitext-103 corpus by $\sim 10 - 16\%$ perplexity gain and $\sim 5 - 8\%$ WER reduction.

- **Downstream Application**: SCAN retriever improves context retrieval in personalized text generation and classification via zero-shot prompting and few-shot fine-tuning of LLMs on LaMP corpus by $7 - 12\%$.

## 2 Related Work

**Language Modeling for Rare Words Prediction**: Earliest works explored the use of LSTM with auxiliary pointer networks to predict rare words and long-term dependencies in language modeling (Merity et al., 2017). Neural cache augmentation (Grave et al., 2017) stored past hidden activations in cache memory to predict out-of-vocabulary words. Implicit cache memorization (Li et al., 2020b) used cache to store past word occurrences as an alternative to the attention-based pointer mechanism. For the ASR re-scoring task, cross-sentence neural LMs proposed to use word usage in preceding sentences to re-rank n-best hypotheses of upcoming sentences (Sun et al., 2021; Irie et al., 2019b).

**Retrieval Augmentation for Language Modeling**: kNN-LM (Khandelwal et al., 2020) interpolated pre-trained LMs with contexts extracted from an external data store using the kNN algorithm. REALM (Guu et al., 2020) proposed a neural retriever to leverage external knowledge during LM pre-training. (Ram et al., 2023) augmented GPT-2 with a large episodic memory for a zero-shot reduction in perplexity. Retrieval-Enhanced Transformer (Retro) (Borgeaud et al., 2022) retrieved document chunks similar to preceding tokens using a cross-attention module. Our work is the first to use domain-distributed n-gram representations over document spans to retrieve rare word patterns from the most relevant external knowledge domains.

## 3 PersonaLM Retriever Augmentation

Fig. 2 describes our proposed PersonaLM retrieval augmentation approach that biases the predictions from a base LM with the next word probabilities based on the relevance of unseen topic/users to the input query. Given an input query $q = (w_1, \cdots, w_{t-1})$ at inference, autoregressive LMs estimate the probability distribution for target token $w_t$ as $P_{LM}(w_t|q)$. To augment the LM output with domain-specific word occurrence information, we calculate the probability distribution of next word prediction over the vocabulary conditioned on the relevance of the underlying domains $(d_1, d_2, \cdots, d_K)$ to the query $(q)$ by marginalizing

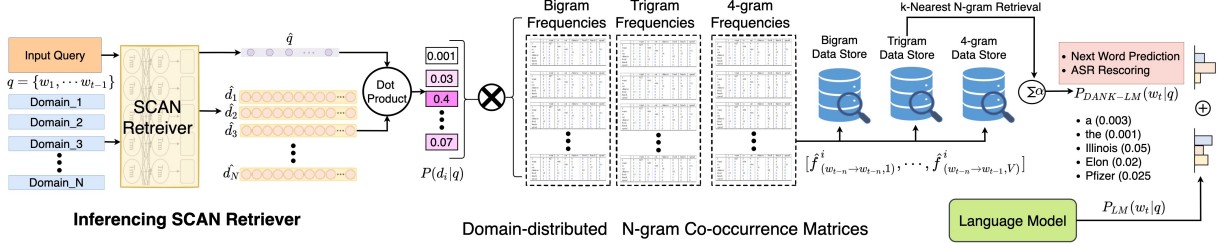

Figure 2: PersonaLM: At inference, we compute the relevance score $P(d_i|q)$ between the query and domains $d_i$ as the dot product of their SCAN retriever representations. We construct a data store for each n-gram frequency matrix. k-most similar n-gram contexts wrt to the input query are retrieved and their weighted summation based on the domain's relevance score is computed to get probability distribution over targets $P_{PersonaLM}(w_t|q)$ and interpolated with LM probabilities $P_{LM}(w_t|q)$.

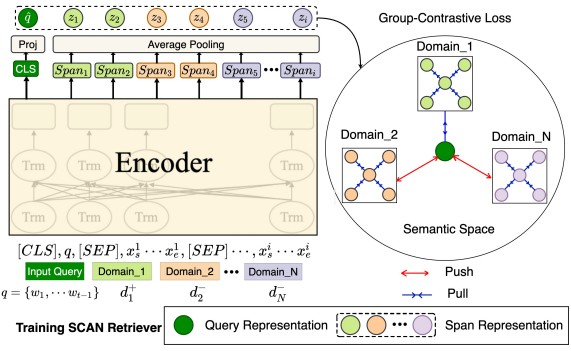

Figure 3: SCAN Retriever: Input query $q$ followed by text spans $(x_s, \cdots, x_e)$ from the positive domain $(d^+)$ and $N-1$ negative domains $(d^-)$ separated by [SEP] are passed through the encoder followed by a projector layer and an average pooling layer. SCAN retriever is trained via a group-wise contrastive loss to force the hidden representation of the query $\hat{q}$ close to its own spans $z_i$, while far away from other groups.

over the retrieved documents as:

$$P_{D-RAG}(w_t|q) = \sum_{i=1}^{K} P(d_i|q) \times P(w_t|d_i, q)$$

### 3.1 SCAN Retriever

Fig. 3 shows the architecture of our proposed Span Aggregated Group-Contrastive Neural (SCAN) retriever which is a Transformer encoder pre-trained with Masked Language Modeling (MLM) as well as a novel group-wise contrastive span loss to force the semantic representations of the input query close to its ground truth domain and away from random spans from the different domains. During the training of the SCAN retriever, we first sample spans of varying granularities from multiple domains and encode them using the Transformer encoder. Group-wise contrastive loss is then applied to learn discriminative semantic presentations for enhanced retrieval performance.

**Document Span Sampling**: Different granularities of spans capture different properties of the input text. For example, phrase-level spans can capture specific words or entities mentioned in the

text while paragraph-level spans can capture more abstract properties of the text such as topic information. In this work, we explicitly sample a set of text spans at the phrase level, sentence level, and paragraph level. We extract $T$ spans for each level of granularity to obtain a total of $3T$ spans corresponding to each input document $D$. Text spans $(x_s, \cdots, x_e)$ are sampled such that their start position is taken from a uniform distribution $U(1, l-1)$ the span length $l = e - s + 1$ is determined by a beta distribution $B(a, b) \times (l - s)$, where $l$ denotes the number of phrases/sentences/paragraphs in the document with $a, b$ as hyperparameters.

**Multi-domain Text Encoding**: Formally, let there be a query $q$ with an associated positive domain $d^+$ and a pool of $N-1$ negative domains $(d_i^-)$. For each domain, we concatenate the input query with multiple spans, add a special [CLS] token before the query text and a [SEP] token between the multiple spans to obtain the concatenated text sequence $t = [CLS], q, [SEP], x_s^1 \cdots x_e^1, [SEP] \cdots, x_s^N \cdots x_e^N$. We encode the input text sequence using a multi-layer Transformer encoder which maps each word to a low-dimensional dense representation $h_0, h_1, \cdots, h_i = Transformer(x_0, x_1, \cdots, x_i)$, where $h_i \in R^H$ with $H$ as the size of hidden dimension. We then pass the encoded representation through a project layer which is a fully-connected layer followed by a non-linear activation $p_i = Tanh(FFN(h_i))$ to prevent representation collapse during contrastive learning. An average pooling operation is applied over the projected word representations to obtain the output representations as $z = AvgPool(p_s \cdots p_e)$.

**Group-wise Contrastive Training**: We use group-wise contrastive learning that incentives for representations of spans in a group sharing the same semantics to be similar while penalizing the rep-

resentations of groups expressing different semantics to be distinguished from each other. It encourages the SCAN retriever to discriminate and score related query-span pairs (from the same domain) higher than unrelated (from different domains) pairs. Given a mini-batch with $N$ domains, the group-wise contrastive loss function $L_{GC}$ is applied over $M = N * (3T + 1)$ spans as:

$$L_{GC} = \frac{-1}{3T} \sum_{i=1}^{N} \sum_{v \in d^+}^{T} \log \frac{\exp(sim(z_i, z_v)/\tau)}{\sum_{j=1}^{M} 1_{i \neq j} \exp(sim(z_i, z_j)/\tau)}$$

where $sim(\cdot)$ refers to the dot product and $\tau$ is the temperature parameter.

## 3.2 Retrieving Relevant Domains

At inference, we encode the concatenated query text with the document spans through the SCAN retriever. The relevance score assigned by the retriever model to a particular domain $d_i$ based on the input query $q$ is denoted by $P(d_i|q)$ is computed via the dot product operation between the [CLS] token embedding ($z_q$) and the average pooled embeddings of the document spans ($z_{d_i}$) as $P(d_i|q) = sim(z_q, z_{d_i})$.

## 3.3 Constructing k-Nearest N-gram Co-occurrence Matrix

We hypothesize that words that occur together in a specific domain have a high chance of triggering during inference. N-gram frequency matrices store co-occurrence information in a matrix such that each cell $(i, j)$ corresponds to the normalized frequency of the observed word $w_i \in V$ following a sequence of $n - 1$ words in a document. To exploit the word-level co-occurrence probabilities in text, we construct the $n$-gram frequencies matrices for each target domain $d_i$ for $n \in [2, 4]$ over the entire vocabulary set $V$ as $f^i_{[(w_{t-n} \to w_{t-1}), w_v]}$. However, n-gram frequencies tend to get sparser with higher values of $n$. Moreover, restricting the query n-grams to only exact matches leads to a loss of information due to ignorance of semantically similar n-grams (Li et al., 2017). To overcome these drawbacks, we utilize k-nearest n-grams. We construct domain-specific n-gram data stores with keys as the pair of Bert vector representation of the n-gram (BERT($w_{t-n} \to w_{t-1}$)) and next word ($w_v$); values as the frequency of co-occurrence in the given domain $f^i_{[(w_{t-n} \to w_{t-1}), w_v]}$. At inference, PersonaLM uses k-NN with Euclidean distance metric to query the datastore for top-k nearest

neighbor n-grams based on their BERT representations. The top-$k$ probability distributions obtained from the n-gram datastore are summed over the entire vocabulary to get $\hat{f}^i_{[(w_{t-n} \to w_{t-1}), w_v]}$. The next word prediction for a selected domain is calculated as a weighted sum of k-nearest bigrams, trigrams, and 4-gram frequencies as $P(w_t|d_i, q) = \sum_{j=2}^{4} \alpha_j * [\hat{f}^i_{[(w_{t-j} \to w_{t-1}), 1]}, \cdots \hat{f}^i_{[(w_{t-j} \to w_{t-1}), V]}]$, where $\alpha_j \in [0, 1]$ are hyperparameters.

## 3.4 LM Augmentation

We compute the `PersonaLM` retrieved next-word probability $P_{D-RAG}(w_t|q)$ by summing the normalized k-nearest n-gram co-occurrence probabilities weighted by the relevance score of the selected domain over the target vocabulary as:

$$P_{D-RAG}(w_t|q) = \sum_{i=0}^{K} sim(z_q, z_{d_i}) \times$$

$$(\sum_{j=2}^{n} \alpha_j * [\hat{f}^i_{[(w_{t-j} \to w_{t-1}), 1]}, \cdots \hat{f}^i_{[(w_{t-j} \to w_{t-1}), V]}])$$

Finally, we interpolate the retrieved next-word probability distribution through PersonaLM ($p_{D-RAG}$) with the base LM output ($P_{LM}$) using a hyperparameter $\lambda$ to produce the final next-word probability distribution as:

$$P(w_t|q) = \lambda P_{D-RAG}(w_t|q) + (1 - \lambda) P_{LM}(w_t|q)$$

# 4 Experiments

## 4.1 Training SCAN retriever

We start with a pre-trained BERT model as the encoder which is further trained using group-wise contrastive learning objective on the following IR benchmarks: (1) MS MARCO Passage Ranking (MARCO Dev Passage), (2) MS MARCO Document Ranking (MARCO Dev Doc) Nguyen et al. (2016); (3) TREC 2019 Passage Ranking (TREC2019 Passage) and (4) TREC 2019 Document Ranking (TREC2019 Document) Craswell et al. (2020). While training the SCAN retriever, we train the Transformer model with permutations of span order. Hence, the spans from positive domains are not necessary to always precede the spans from negative domains. Similarly, the spans from an individual domain are not necessitated to be always together in the sequence. This ensures the learned span representations are robust in terms of their relative order. More details are in Appendix. B.9.

## 4.2 Datasets

We evaluate the PersonaLM method on our proposed ASAP dataset and the UserLibri corpus.

We propose the **ASR Language Model Personalization (ASAP)** benchmark which aims to evaluate the efficacy of LMs for personalized automatic speech recognition based on user/domain-specific information for next word prediction and ASR n-best second-pass restoring.

**(1) Personalized Meeting ASR** of user's spoken utterances in professional meetings. This task assesses the model's ability to capture user-preferred dialogue patterns and linguistic characteristics. We leverage the AMI Meeting corpus (Kraaij et al., 2005) by splitting the user-specific recordings to obtain personalized utterance-text pairs.

**(2) Personalized TED Talk ASR** to convert recorded TED talks delivered by a specific user into text transcript. This task evaluates the LM's capability to capture topics-aware word patterns in the speeches from the TED-LIUM v3 corpus (Hernandez et al., 2018). We split the recorded TED talks temporally with historical utterance-text pairs forming the domain-specific train set.

**(3) Personalized Financial Earning Calls ASR**: Perform speech-to-text for financial earnings conference calls that include company-specific financial information. The task aims to evaluate a language model's capacity to extract company-specific named entities, abbreviations, facts, and long-tail word patterns. We adopt the conference call-transcript pairs from combined Earnings-21 (Rio et al., 2021) and Earnings-22 (Del Rio et al., 2022) datasets. Additionally, we use Wikitext-103 (Merity et al., 2017) to test the LM domain adaptation in topic-specific documents. PersonaLM can be applied to a variety of different datasets with varying "domains" i.e. speakers/topics/categories. For example, the recordings in the AMI Meeting corpus are split based on speaker IDs. Domains in Earnings-21+22 data correspond to different companies. A specific domain in the TED-LIUM v3 dataset refers to a TED talk speaker. Domains in Wikitext-103 correspond to individual Wikipedia pages. **Data Preprocessing**: To study personalization in language modeling, we reformulated all the listed datasets to identify explicit users/domains. For each dataset, we combined the original train/val/test portions and splitted user-based data in the ratio of 70:10:20 such that each user/domain appears only in one of splits. Table

| Dataset | Train | Val | Test | Vocab Size | Domain | # Domains |
|---|---|---|---|---|---|---|
| Earnings-21+22 | 49.6K | 7.1K | 14.2K | 20K | Financial Company | 169 |
| AMI Meeting Corpus | 17.1K | 2.7K | 5.8K | 11K | Meeting Speaker | 135 |
| TED-LIUM v3 | 188.9K | 26.6K | 9.3K | 46K | TED speaker | 2351 |
| Wikitext-103 | 2M | 300K | 10K | 200K | Wikipedia Page | 30k |
| UserLibri | 6.3M | 700K | 10K | 10K | Books | 107 |

Table 1: Data stats of ASAP, UserLibri, and WikiText-103.

1 shows statistics on dataset size and distribution. **UserLibri Dataset** (Breiner et al., 2022a) reformulates the Librispeech corpus into user-specific audio-transcript pairs supplemented with personalized text-only data corresponding to each user with similar vocabulary, character names, and writing styles as the recordings.

## 4.3 Experiments for ASR Personalization

**Language Model Architecture**: We experiment with both LSTM and Transformer LMs. LSTM model has 2 layers with a 300-d embedding layer and a hidden dimension of 1500. Transformer LM consists of 4 layers of encoder-decoder with 12 heads, 128-d hidden representations, and a feed-forward layer of 3072-d. For generating ASR n-best hypotheses, we use a pre-trained RNN-T ASR Model with Emformer encoder (Shi et al., 2021), LSTM predictor, and a joiner with 80M parameters. More hyperparameter details in Appendix Sec.B.9

**Pre-training LMs**: LSTM and Transformer LMs are pre-trained on Librispeech (Panayotov et al., 2015) train set for 25 epochs with batch size of 256, Adam optimizer and cross-entropy loss for next word prediction. We select model checkpoints with least perplexity on the Librispeech validation.

**Adaptation to Unseen Domains**: We evaluate the retrieval augmentation in two settings: (1) *Without fine-tuning*: LM pre-trained on generic corpus; (2) *With fine-tuning*: LM pre-trained on generic corpus and fine-tuned on the entire out-of-domain train corpus. In both cases, evaluation is performed on the out-of-domain test set.

**Baselines**: **(i) LSTM/Transformer**: Language model without any augmentation, **(ii) Neural Cache Model** (Grave et al., 2017) augments LM output with continuous a cache memory of previous hidden states. The stored keys are used to retrieve the next word through a dot product-based memory lookup with the query. **(iii) kNN-LM** (Khandelwal et al., 2020): Following (Das et al., 2022), we adopt kNN-LM to memorize context vectors from representations from out-of-domain train set in an external data store. During inference, the k-nearest neighbors of the decoder output representations are

interpolated with LM output. (iv) **Unified N-gram Co-occurrence**: N-gram word frequency matrices built from the combined out-of-domain train set of each user/domain are augmented with LM at inference. (v) **PersonaLM** $w\backslash$ **other retrievers**: Replacing SCAN retriever with DPR (Karpukhin et al., 2020) or Contriever (Izacard et al., 2021).

**Ablation Studies**: **(i) PersonaLM w\o SCAN retriever**: We use the dot product of query and domain context encoded through pre-trained BERT to compute the weightage of each domain; **(ii) PersonaLM w\o k-Nearest N-grams**: Similar to kNN-LM, we augment the NWP with k-nearest Bert contexts vectors extracted from individual domains. We compute relevance scores from the SCAN retriever to get the weighted sum of the probability distributions from each domain.

**Evaluation Metrics**: We utilize word-level perplexity scores to evaluate LM performance for next-word prediction. We also report Word Error Rate (WER) for ASR second-pass re-scoring for ASAP corpus. For UserLibri, we evaluate WER per user for both streaming and non-streaming ASR settings. For each model, we report results with minimal perplexity by iterating the interpolation parameter $\lambda$ between 0 to 1 in increments of 0.1.

**ASR Model Architecture for UserLibri**: We utilize separate architectures for streaming and non-streaming ASR. The Conformer Hybrid Autoregressive Transducer (HAT) from Breiner et al. (2022a) has 86M parameter and consists of 12 encoder layers of 512-d, 4 attention heads, convolution kernel size of 32, and a HAT decoder with a single RNN layer of 640-d. Each label is embedded with 128-d, and inputs are tokenized with a 1k Word-Piece Model trained on the LibriSpeech train set. The models are trained with Adam using group-norm (Wu and He, 2018). For streaming ASR evaluation, the Conformer HAT model uses causal convolution, local self-attention, and left-sided context stacking to ensure no look-ahead. The non-streaming version has multi-headed attention. COnformer models in both cases are trained on 960 hours of LibriSpeech audio training set. The LSTM decoder in streaming ASR is a 2-layer RNN of size 1340 with 25M parameters. It uses a similar Word Piece model as the Conformer.

## 4.4 SCAN Retriever Experiments on LaMP

**LaMP** (Salemi et al., 2023) is a benchmark corpus to evaluate LM personalization on the follow-

| | Model | WikiText-103 Perplexity ($\downarrow$) | Earnings-21+22 Perplexity ($\downarrow$) |
|---|---|---|---|
| Without Fine-tuning | LSTM | 1384.1 | 757.6 |
| | + Neural Cache | 1325.3 | 723.8 |
| | + kNN-LM | 1191.6 | 659.1 |
| | + Unified N-gram Co-occurrence Retrieval | 603.6 | 477.2 |
| | + PersonaLM w \DPR | 544.3 | 420.3 |
| | + PersonaLM w \Contriever | 539.8 | 412.3 |
| | + PersonaLM | **527.9** | **405.6** |
| | + PersonaLM w\o SCAN Retriever | 542.8 | 415.4 |
| | + PersonaLM w\o k-Nearest N-gram | 542.3 | 415.7 |
| With Fine-tuning | LSTM | 103.9 | 66.2 |
| | + Neural Cache | 97.6 | 66.0 |
| | + kNN-LM | 91.8 | 65.7 |
| | + Unified N-gram Co-occurrence Retrieval | 89.2 | 64.5 |
| | + PersonaLM w \DPR | 84.2 | 63.0 |
| | + PersonaLM w \Contriever | 80.2 | 59.6 |
| | + PersonaLM | **77.8** | **57.8** |
| | + PersonaLM w\o SCAN Retriever | 82.7 | 61.5 |
| | + PersonaLM w\o k-Nearest N-gram | 82.3 | 61.9 |

(a) LSTM LM

| | Model | WikiText-103 Perplexity ($\downarrow$) | Earnings-21+22 Perplexity ($\downarrow$) |
|---|---|---|---|
| Without Fine-tuning | Transformer | 1322.3 | 834.2 |
| | + Neural Cache | 1295.3 | 802.4 |
| | + kNN-LM | 1150.4 | 717.8 |
| | + Unified N-gram Co-occurrence Retrieval | 585.3 | 454.8 |
| | + PersonaLM w \DPR | 578.2 | 452.7 |
| | + PersonaLM w \Contriever | 569.3 | 446.1 |
| | + PersonaLM | **567.6*** | **440.4*** |
| | + PersonaLM w\o SCAN Retriever | 572.9 | 448.2 |
| | + PersonaLM w\o k-Nearest N-gram | 572.1 | 447.9 |
| With Fine-tuning | Transformer | 88.6 | 55.2 |
| | + Neural Cache | 86.8 | 54.9 |
| | + kNN-LM | 79.3 | 54.2 |
| | + Unified N-gram Co-occurrence Retrieval | 76.5 | 53.8 |
| | + PersonaLM w \DPR | 76.1 | 52.6 |
| | + PersonaLM w \Contriever | 72.5 | 49.6 |
| | + PersonaLM | **70.9*** | **48.2*** |
| | + PersonaLM w\o SCAN Retriever | 74.8 | 51.5 |
| | + PersonaLM w\o k-Nearest N-gram | 73.5 | 50.3 |

(b) Transformer LM

Table 2: Results comparing the performance of `PersonaLM` Retrieval Augmentation for (a) LSTM and (b) Transformer LMs with baselines and ablations (in red) for the **Next Word Prediction** task on WikiText-103 and Earnings-21+22 datasets. PersonaLM achieves the lowest perplexity scores across all settings. * indicates that the result is statistically significant (5 runs) based on Wilcoxon's signed rank test ($p < 0.001$).

ing user-specific text classification and generation tasks: (1) citation identification, (2) news categorization (3) product rating prediction, (4) news headline generation, (5) scholarly title generation. Each data sample contains an input sequence to the model, a target output, and several text samples that encapsulate the user profiles that can be employed for LLM personalization.

**Baselines**: Inspired by (Salemi et al., 2023), we compare SCAN retriever with strong baseline retrievers for user-specific context selection: 1) Random, (2) BM25, and (3) Contriever.

**Evaluation on LLM Personalization**: We evaluate different retrievers for personalized prompt construction in following settings: (a) Zero-shot LLM prompting: Retrieve top-k most relevant user items from external corpus to append in prompts for GPT-

**(a) LSTM LM**

| Model | AMI Meeting Corpus | | TED LIUMv3 | |
|---|---|---|---|---|
| | Perplexity (↓) | WER (↓) | Perplexity (↓) | WER (↓) |
| **Without Fine-tuning** | | | | |
| Audio Model Only (Emformer) | – | 32.54 | – | 17.23 |
| Audio Model + LSTM | 1636.4 | 31.75 | 427.7 | 13.51 |
| + Neural Cache | 1545.4 | 31.69 | 414.5 | 13.25 |
| + kNN-LM | 1232.2 | 31.62 | 389.7 | 7.82 |
| + Unified N-gram Co-occurrence Retrieval | 606.7 | 31.25 | 335.4 | 7.34 |
| + PersonaLM w \DPR | 490.5 | 31.22 | 332.8 | 7.23 |
| + PersonaLM w \Contriever | 471.2 | 31.15 | 315.0 | 7.16 |
| + PersonaLM | 463.8* | 31.01* | 313.8* | 7.01* |
| + PersonaLM w\o SCAN Retriever | 480.2 | 31.13 | 320.3 | 7.15 |
| + PersonaLM w\o k-Nearest N-gram | 478.9 | 31.10 | 318.8 | 7.14 |
| **With Fine-tuning** | | | | |
| Audio Model Only (Emformer) | – | 32.54 | – | 17.23 |
| Audio Model + LSTM | 37.7 | 31.40 | 132.6 | 13.13 |
| + Neural Cache | 37.5 | 31.36 | 132.2 | 13.03 |
| + kNN-LM | 37.1 | 31.27 | 131.5 | 7.76 |
| + Unified N-gram Co-occurrence Retrieval | 36.6 | 31.20 | 130.3 | 7.44 |
| + PersonaLM w \DPR | 36.2 | 31.16 | 130.2 | 7.28 |
| + PersonaLM w \Contriever | 35.1 | 31.14 | 128.7 | 7.03 |
| + PersonaLM | 34.7* | 31.01* | 127.5* | 6.90* |
| + PersonaLM w\o SCAN Retriever | 35.9 | 31.14 | 129.7 | 7.10 |
| + PersonaLM w\o k-Nearest N-gram | 35.7 | 31.12 | 129.4 | 7.07 |

**(b) Transformer LM**

| Model | AMI Meeting Corpus | | TED LIUMv3 | |
|---|---|---|---|---|
| | Perplexity (↓) | WER (↓) | Perplexity (↓) | WER (↓) |
| **Without Fine-tuning** | | | | |
| Audio Model Only (Emformer) | – | 32.54 | – | 17.23 |
| Audio Model + Transformer | 2114.3 | 32.05 | 442.0 | 13.24 |
| + Neural Cache | 1987.5 | 32.01 | 424.5 | 13.18 |
| + kNN-LM | 1579.0 | 31.95 | 398.6 | 7.57 |
| + Unified N-gram Co-occurrence Retrieval | 637.1 | 31.37 | 332.3 | 7.22 |
| + PersonaLM w \DPR | 624.9 | 31.33 | 327.4 | 7.14 |
| + PersonaLM w \Contriever | 601.4 | 31.25 | 310.1 | 7.05 |
| + PersonaLM | 592.6* | 31.16* | 308.3* | 6.92* |
| + PersonaLM w\o SCAN Retriever | 610.3 | 31.29 | 316.6 | 7.15 |
| + PersonaLM w\o k-Nearest N-gram | 608.5 | 31.27 | 315.5 | 7.05 |
| **With Fine-tuning** | | | | |
| Audio Model Only (Emformer) | – | 32.54 | – | 17.23 |
| Audio Model + Transformer | 29.5 | 31.28 | 116.7 | 12.98 |
| + Neural Cache | 29.3 | 31.24 | 116.2 | 12.78 |
| + kNN-LM | 29.1 | 31.19 | 115.6 | 7.35 |
| + Unified N-gram Co-occurrence Retrieval | 28.1 | 31.14 | 114.0 | 7.21 |
| + PersonaLM w \DPR | 27.7 | 31.10 | 113.0 | 7.04 |
| + PersonaLM w \Contriever | 26.4 | 31.03 | 112.3 | 6.93 |
| + PersonaLM | 25.7* | 30.88* | 111.1* | 6.86* |
| + PersonaLM w\o SCAN Retriever | 27.0 | 31.09 | 112.7 | 7.00 |
| + PersonaLM w\o k-Nearest N-gram | 26.9 | 31.05 | 112.8 | 6.98 |

Table 3: Results comparing the performance of `PersonaLM` Retrieval Augmentation for (a) LSTM and (b) Transformer LMs with baselines and ablations (in red) for the **Next Word Prediction** and **Second-Pass ASR Re-scoring** tasks on AMI Meeting Corpus and TED LIUMv3 datasets. PersonaLM achieves minimum perplexity WER on both datasets. * indicates that the result is statistically significant (5 runs) based on Wilcoxon's signed rank test ($p < 0.001$).

| Model | Streaming | | | Non-Streaming | | |
|---|---|---|---|---|---|---|
| | Test-Clean | Test-Other | All | Test-Clean | Test-Other | All |
| Conformer Transducer (Audio Model Only) | 6.0 | 11.2 | 8.5 | 2.5 | 6.8 | 4.5 |
| Conformer Transducer + LM (25M) | 5.2 | 9.1 | 7.1 | 2.0 | 5.5 | 3.7 |
| + Fine-tuned LM (p13n) | 5.2 | 8.7 | 6.9 | 1.9 | 4.6 | 3.2 |
| + Unified N-gram Retrieval | 5.1 | 8.6 | 6.8 | 1.9 | 4.4 | 3.1 |
| + PersonaLM $w \setminus$ Contriver | 5.0 | 8.5 | 6.8 | 1.8 | 4.4 | 3.0 |
| + PersonaLM | 4.8* | 8.3* | 6.6* | 1.6* | 4.2* | 2.8* |
| + PersonaLM w/o SCAN retriever | 5.1 | 8.6 | 6.9 | 1.8 | 4.6 | 3.0 |
| + PersonaLM w/o k-Nearest N-gram | 5.0 | 8.5 | 6.8 | 1.8 | 4.5 | 3.1 |

Table 4: Performance comparison of `PersonaLM` Retrieval Augmentation with baselines and ablations for personalized (a) streaming ASR and (b) non-streaming ASR on the UserLibri dataset. PersonaLM reduces the WER by 7-12% across all settings. * indicates that the result is statistically significant (5 runs) based on Wilcoxon's signed rank test ($p < 0.001$).

3.5[1] and FlanT5-XXL (Chung et al., 2022); (b) Few-shot LM Fine-tuning: Fine-tuning FlanT5-base (Chung et al., 2022) using top-k retrieved items from the user profile. More details on fine-tuning FlanT5-base in Appendix Sec. B.9

## 5 Results and Analysis

**Perplexity Evaluation**: Tables 2 and 3 compare the perplexity scores of the proposed `PersonaLM` retrieval augmentation against other baselines. We observe that the Neural Cache model (Li et al., 2020b) slightly improves over naive LM baselines but struggles due to its inability to handle long-range dependencies through pointer mechanism. Consistent with observations of Wang et al. (2023), kNN-LM (Khandelwal et al., 2020) reduces perplexity by 5-10% but is still challenged by the non-parametric fuzzy nature of k-nearest Bert context

---
[1] https://platform.openai.com/docs/models/gpt-3-5

vectors selected amongst billions of stored contexts from a gigantic data store. Similar to (Drozdov et al., 2022), our experiments show that Unified N-gram Co-occurrence shows slight improvement over kNN-LM as n-grams are better at capturing highly domain-specific rare word patterns. However, it still suffers from sub-optimal n-gram retrievals from a mixture of domains as the target probabilities get averaged out when computed over the entire external corpus. Our proposed method achieves SOTA performance and improves the LM perplexity by a significant margin on WikiText-103 ($57.1 - 61.8\%$ w\o fine-tuning, $20.0 - 25.1\%$ with fine-tuning), Earnings21+22 ($46.4 - 47.2\%$ w\o fine-tuning, $11.4 - 12.6\%$ with fine-tuning), AMI Meeting Corpus ($71.6 - 71.9\%$ w\o fine-tuning, $7.9 - 12.8\%$ with fine-tuning), and TED LIUMv3 ($26.6 - 30.2\%$ w\o fine-tuning, $3.8 - 4.7\%$ with fine-tuning) over base LMs, demonstrating that contextually matching query with most relevant domains via SCAN retriever module boosts retrieval performance which reinforces the next word prediction task. Replacing the SCAN retriever in PersonaLM with other baseline retrievers like Dense Passage Retriever (DPR) or Contriever leads to degraded performance. However, the performance does not decrease below kNN-LM or Unified N-gram Co-occurrence Retrieval methods, signifying the marginal benefit of domain-specific retrieval to augment LM predictions.

**ASR Rescoring Analysis on ASAP dataset**: Table 3 shows results of second-pass ASR rescoring on AMI Meetings and TED LIUMv3 datasets where our proposed approach improves WER relatively by $\sim 5\%$. Retrieval-augmented LMs when combined with the n-best hypotheses produced by the audio model lead to statistically significant WER reduction with respect to both kNN-LM and PersonaLM with Contriever baselines. Combining audio model and PersonaLM allows wins on tail

| Dataset | Metric | FlanT5-XXL | | | GPT-3.5 | | | FlanT5-base (fine-tuned) | | |
|---|---|---|---|---|---|---|---|---|---|---|
| | | Non-personalized | Contriver | SCAN Retriever | Non-personalized | Contriver | SCAN Retriever | Non-personalized | Contriver | SCAN Retriever |
| LaMP-1U: Personalized Citation Identification | Accuracy | 0.522 | 0.675 | **0.687** | 0.510 | 0.701 | **0.715** | 0.522 | 0.731 | **0.745** |
| LaMP-2U: Personalized News Categorization | Accuracy | 0.591 | 0.598 | **0.608** | 0.610 | 0.693 | **0.702** | 0.730 | 0.835 | **0.843** |
| | F1 | 0.463 | 0.471 | **0.484** | 0.455 | 0.455 | **0.466** | 0.504 | 0.637 | **0.648** |
| LaMP-3U: Personalized Product Rating | MAE | 0.357 | 0.282 | **0.276** | 0.699 | 0.658 | **0.644** | 0.314 | 0.258 | **0.246** |
| | RMSE | 0.666 | 0.584 | **0.565** | 0.977 | 1.102 | **0.980** | 0.624 | 0.572 | **0.559** |
| LaMP-4U: Personalized News Headline Generation | ROUGE-1 | 0.164 | 0.192 | **0.211** | 0.133 | 0.160 | **0.172** | 0.158 | 0.201 | **0.212** |
| | ROUGE-L | 0.149 | 0.178 | **0.187** | 0.118 | 0.142 | **0.155** | 0.144 | 0.185 | **0.192** |
| LaMP-5U: Personalized Scholarly Title Generation | ROUGE-1 | 0.455 | 0.467 | **0.475** | 0.395 | 0.398 | **0.409** | 0.424 | 0.453 | **0.470** |
| | ROUGE-L | 0.410 | 0.424 | **0.433** | 0.334 | 0.336 | **0.342** | 0.382 | 0.414 | **0.425** |

Table 5: Performance comparison of zero-shot FlanT5-XXL, GPT-3.5, and few-shot fine-tuned FlanT5-base for personalized text classification and generation results on the eval set of LaMP dataset. For all metrics the higher the better, except for RMSE and MAE. Prompting LLMs with user-specific context selected by the SCAN retriever consistently reports the best performance.

words while avoiding losses on common word occurrences. **Evaluation on UserLibri dataset** (Breiner et al., 2022b) shows that PersonaLM improves WER for both streaming and non-streaming ASR. Compared to fine-tuning the LM on the entire external personalized corpus (p13n LM), PersonaLM can selectively learn user-specific discriminative patterns in speech text and weigh it appropriately for biasing the LM predictions.

**Ablation Analysis**: Tables 2-4 highlights in red show the ablation study for PersonaLM. We observe that SCAN retriever is critical in all settings due to its enhanced ability to learn enhanced discriminative document representations that help assign appropriate weights to external domains. Removing the k-nearest N-grams severely deteriorates the performance as the LM is no longer able to exploit the personalized n-gram probability distribution from different domains. The severe performance drop in WER for speech datasets in the absence of either of the components underscores their significance for personalized ASR tasks.

**Adaptation to Unseen Domains**: Retrieval augmentation with fine-tuned LMs shows a sustained relative gain of 5-18% across all settings despite having seen the same data during the fine-tuning stage. This observation validates our hypothesis that despite the benefits of transfer learning for out-of-domain generalization, explicit memorization is needed to effectively learn user-specific linguistic patterns not retained during fine-tuning.

**Qualitative Examples**: Qualitative examples from ASAP, Wikitext, and UserLibri datasets in Table 6 along with model predictions. PersonaLM is able to able to correctly predict proper nouns, abbreviations, and homonyms mistaken by fine-tuned LM and kNN-LM baselines, while also fixing the problem of over-prediction of domain-specific frequent words commonly observed in Unified N-gram Co-occurrence Retrieval baseline.

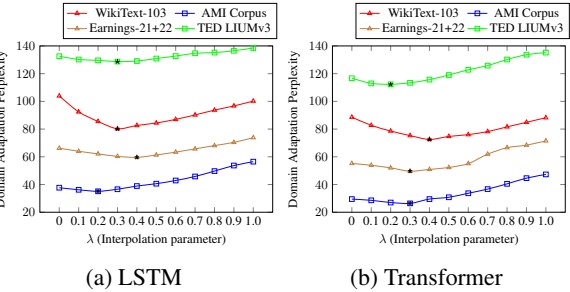

(a) LSTM      (b) Transformer

Figure 4: Plot of $\lambda$ (interpolation parameter) vs perplexity of PersonaLM with fine-tuned (a) LSTM, (b) Transformer LMs on WikiText-103, Earnings-21+22, AMI Corpus, and TED LIUMv3 datasets. Curves show convex characteristics with the optimal value of $\lambda$ varying with different settings.

**Impact of Interpolation Parameter**: Figure 4 shows the optimal value of interpolation parameter $\lambda$ varies in different settings. $\lambda$ vs perplexity curve shows convex characteristics for all variants of PersonaLM. Perplexity scores improve with increasing $\lambda$ as explicit memorization of rare word patterns mined from matching domains benefits the next word prediction task but starts to drop monotonically after reaching an inflection point.

**Downstream Application of SCAN Retriever**: Table 5 shows the application of different retrievers for improving user-personalized text generation and classification on the LaMP dataset. We aim to retrieve the most relevant user profiles that can be augmented with query context for zero-shot prompting or few-shot fine-tuning of LLMs. SCAN retriever outperforms both BM-25 and Contriever baselines and shows significant gains across different metrics compared to non-personalized LMs across all subtasks. As opposed to earlier advances where retrieval augmentation was performed during LM training (Lewis et al., 2020; Guu et al., 2020), the core merits of our proposed SCAN retriever are that it is extensible to any LM (LSTM, Transformer, GPT, etc), can seamlessly adapt to new users, enable in-context retrieval augmenta-

| Win/Loss | Ground Truth | Fine-tuned LM | kNN-LM | Unified N-gram Retrieval | PersonaLM | Type |
|---|---|---|---|---|---|---|
| Win | king sharr khan | king sir can | king sir khan | king share can | kingsharr khan | Proper Name |
| Win | murdoch blinked | mr duck winged | mom duck blink | murdock blinked | murdock blinked | Proper Name |
| Loss | tied to a woman | tied to a woman | tied tooa woman | died to a woman | died to a woman | homonym |
| Win | lord of baghdad | lord of bag dad | lord of bag dad | lord baghdad | lord of baghdad | homonym |
| Win | mister beale | mister bell | mister bell | mister be elle | mister beale | Proper Name |
| Win | thanks izzy | thanks is he | thanks is he | thank is he | thanks izzy | homonym |
| Win | North American Treaty Alliance | North American Treaty All Hands | North American Treaty Organization | North American Treaty Alliance | North American Treaty Alliance | Abbreviation Term |
| Loss | utterly RSVP for this our invitation | utter RSVP for this our invitation | utter respect for this our invitation | utterly rest for this our invitation | utterly respectively for this our invitation | Abbreviation Term |

Table 6: Qualitative examples: Ground truth, baseline predictions, and PersonaLM predictions for a few samples from UserLibri and ASAP eval set. PersonaLM is able to able to correctly predict proper nouns, abbreviations, and homonyms mistaken by fine-tuned LM and kNN-LM baselines, while also fixing the problem of over-prediction of domain-specific frequent words commonly observed in Unified N-gram Co-occurrence Retrieval baseline.

tion without any LM-specific fine-tuning, and requires very small memory footprint with negligible computational overhead.

## 6 Conclusion

We introduce PersonaLM retrieval augmentation for ASR personalization using a SCAN retriever trained via group-contrastive learning to rank textual documents from an external knowledge corpus based on their semantic similarity with the input query. We aggregate the probability distribution of the next word prediction by utilizing domain-specific n-gram word frequency representations weighted by the relative importance of the external domains to the input query. Experiments on our proposed ASAP benchmark and the UserLibri dataset show that our method achieves SOTA perplexity and WER. We show that the SCAN retriever is also useful for in-context LLM augmentation for zero-shot prompting and few-shot fine-tuning.

## 7 Limitations

Through careful analysis of error cases, we found that there are two main types of prediction errors from the proposed model. First, our methodology is challenged by the phenomenon of contradictory information. As retrieval augmentation relies on external data that may be from varying sources, fact verification and rare word confounders in certain cases can be confusing for our models. Our approach does not reconcile facts, names or out-of-domain patterns from varying information sources, but rather takes a frequentist approach due to the reliance on n-grams. For example, consider this query - "Nigerian authorities report fresh aggression on June 15, 2018, that led to the death of - ". To complete the query, several external new article sources are ranked highly by our SCAN retriever. However, only a few of them are relevant to the date the query was executed, while some of them were later debunked as misinformation. Tackling this type of error requires fact-checking and document

grounding mechanisms to cite what information was used for next-word prediction. Another major limitation of this method is that it ignores accented speech and speech that causes cold start problems for noisy speech samples wherein the contextual query is corrupted from the start. As a result, ASR rescoring is unable to take advantage of previously predicted utterances. This leads to exacerbating the ASR performance that was already suffering due to noise in the audio domain. For instance, the Conformer model personalized using PersonaLM for TED talks recorded in an auditorium does not generalize well to political speeches conducted in open settings with background crowd noises.

## 8 Ethics Statement

We utilize the publicly available Wikitext, AMI Meetings corpus, TED-LIUM v3, Earnings-21, Earnings-22, UserLibri, and LaMP datasets for ASR and language modeling. Our curated benchmarks are composed of open-source and publicly available datasets repurposed for ASR and LM. Our proposed ASAP benchmark does not provide any new annotations but rather reformulates the data splits for our experiments. Our study on user or domain personalization does not target any known individual, race, gender, topic, or ethnicity. Personalization is limited to a user's linguistic characteristics. We do not utilize any PII at any step in our experiments.

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

## A Appendix

### A.1 Limitations

Through careful analysis of error cases, we found that there are two main types of prediction errors from the proposed model. First, our methodology is challenged by the phenomenon of contradictory information. As retrieval augmentation relies on external data that may be from varying sources, fact verification and rare word confounders in certain cases can be confusing for our models. Our approach does not reconcile facts, names or out-of-domain patterns from varying information sources, but rather takes a frequentist approach due to the reliance on n-grams. For example, consider this query - "Nigerian authorities report fresh aggression on June 15, 2018, that led to the death of - ". To complete the query, several external new article sources are ranked highly by our SCAN retriever. However, only a few of them are relevant to the date the query was executed, while some of them were later debunked as misinformation. Tackling this type of error requires fact-checking and document grounding mechanisms to cite what information was used for next-word prediction. Another major limitation of this method is that it ignores accented speech and speech that causes cold start problems for noisy speech samples wherein the contextual query is corrupted from the start. As a result, ASR rescoring is unable to take advantage of previously predicted utterances. This leads to exacerbating the ASR performance that was already suffering due to noise in the audio domain. For instance, the Conformer model personalized using PersonaLM for TED talks recorded in an auditorium does not generalize well to political speeches conducted in open settings with background crowd noises.

**Potential Risks:** Our models are exploratory and academic in nature and should not be used for real-world financial/political/healthcare purposes without extensive investigations into their shortcomings/randomness/biases. **Unhandled Cases**: The current work is limited to the English language and would need suitable tools in other languages to process audio, LM tokenization, vocabulary, and pre-trained multilingual language models. Moreover, our method has been tested on limited domains of Wikipedia text, professional meetings, recorded speeches, and financial earning calls. Applications to life-critical scenarios such as healthcare, public safety, and law will need further investigation.

## B Reproducibility Checklist

### B.1 Summary of paper's main claims

The novel contributions of our work include:

- **PersonaLM** retrieval augmentation for ASR personalization that leverages group-wise contrastive loss to train **Span Aggregated Group-Contrastive Neural (SCAN) retriever** for ranking query-relevant external domains/users and augments domain-distributed k-nearest n-gram frequencies to improve LM predictions.
- **ASAP - a novel benchmark for ASR LM personalization** consisting of three user-specific ASR tasks in the domains of meetings, TED talks, and financial conference calls. PersonaLM significantly outperforms strong baselines on ASAP benchmark, UserLibri, and Wikitext-103 corpus by $\sim 10 - 16\%$ perplexity gain and $\sim 5 - 8\%$ WER reduction.
- **Downstream Application**: SCAN retriever improves context retrieval in personalized text generation and classification via zero-shot prompting and few-shot fine-tuning of LLMs on LaMP corpus by $7 - 12\%$.

### B.2 Citation to creators of artifacts

We use four publicly available datasets for evaluation - WikiText-103 (Merity et al., 2017), combined Earnings-21 + Earnings-22 (Rio et al., 2021; Del Rio et al., 2022), AMI Meeting (Kraaij et al., 2005), and TED-LIUM v3 (Hernandez et al., 2018). We use Librispeech (Panayotov et al., 2015) text corpus for LM pre-training. We prefer to combine both versions of Earnings Call datasets to make the corpus more domain-diverse. We use the UserLibri corpus to evaluate retrieval augmentation for personalizing streaming and non-streaming ASR. We use the LaMP (Salemi et al., 2023) dataset for personalized text generation and classification through retrieval augmentation with LLMs.

- WikiText-103 Dataset[2]

- Earnings-21 Datatset[3]

- Earnings-22 Dataset[4]

- AMI Meeting Corpus[5]

---

[2]https://blog.salesforceairesearch.com/the-wikitext-long-term-dependency-language-modeling-dataset/

[3]https://github.com/revdotcom/speech-datasets/tree/main/earnings21

[4]https://github.com/revdotcom/speech-datasets/tree/master/earnings22.

[5]https://groups.inf.ed.ac.uk/ami/corpus/

- TED-LIUM v3 Dataset[6]

- Librispeech Dataset[7]

- userLibri Dataset[8]

- LaMP Dataset[9]

## B.3 License and terms for use of data artifacts

All the datasets are publicly available and free to use for research purposes.

## B.4 Intended use of data artifacts:

The intended use of text and speech datasets is to improve language modeling and ASR re-scoring. Enhancements in speech recognition and general NLP models can increase the accessibility of AI tools and open new pathways to deploy AI for social good.

## B.5 Steps taken to protect/anonymize names, identities of individual people, or offensive content:

We do not use any identifiable user data for any experiments. All datasets already come with PII data redacted prior to their public release.

## B.6 Coverage of domains, languages, linguistic phenomena, demographic groups represented in data:

Our work performs next word prediction and ASR second pass re-scoring on data from Wikipedia, financial earning calls, meeting recordings, and speeches in the English language. Adaptation to other languages may need appropriate processing.

## B.7 Data statistics and Processing

The data statistics are given in Table 7. For each text sentence/utterance, all alphabets were lower-case, numbers were converted into their respective word forms, and punctuation was removed. To experiment with the out-of-domain generalization ability of the proposed approach, we reformulated existing datasets to identify explicit domains. For each dataset, we combined the train/val/test portions and considered all sentences/utterances in each Wikipedia page/call/recording as part of a particular domain. We then split sentences within the same domain in the ratio of 70:10:20 to form new train/val/test splits.

---

[6]https://www.openslr.org/51/
[7]https://huggingface.co/datasets/librispeech_asr
[8]https://www.kaggle.com/datasets/google/userlibri
[9]https://lamp-benchmark.github.io/

## B.8 Total computational budget and computing infrastructure:

We performed training and inference of the models on industrial strength CPU and clusters of multiple 32GB V100 GPUs. The model takes between 12-36 hours to train on either of the four datasets. Inference/ Fine-tuning time varies from 3-7 hours per dataset.

## B.9 Experimental setup, hyperparameter search, and best-found values:

Hyper-parameters for our experiments were tuned on the respective out-of-distribution evaluation sets to find the best configurations for different datasets. We detail the training setup as follows:

**Training Setup for PersonaLM Retriever**: We experiment with a range of hyperparameters in `PersonaLM` such as: size of hidden layers in Fully Connected Layer $\{64, 128, 256\}$, dropout $\delta \in \{0.1, 0.2, 0.3, 0.4, 0.5.0.6\}$, learning rate $\lambda \in \{1e--5, 2e-5, 3e-5, 4e-5, 5e-5\}$, weight decay $\omega \in \{1e-6, 1e-5, 1e-4, 1e-3\}$, batch size $b \in \{4, 8, 16, 32, 64, 128, 256, 512\}$ and epochs ($\leq 25$). We use a pre-trained BERT model and further train it using contrastive learning loss.

**Pre-training Language Models**: For all experimental settings, we use an LSTM model of 2 layers with a 300-dimension embedding layer and a hidden dimension of 768. The Transformer model consists of 4 layers of encoder-decoder with 12 heads, 128 dimensions hidden representations, Layer Norm input dimensions of 768, attention dropout of 0.2, and a feed-forward layer of 3072 dimensions. We use a batch size of 256 and train till 25 epochs. We use a learning rate of $1e-4$, a fixed learning rate scheduler, a force anneal of 26, a learning rate shrink of 0.8, and a Gelu activation function. We use Adam optimizer with eps of $1e-8$, weight decay of $1e-6$, and betas of $(0.9, 0.999)$ to compute cross-entropy loss as word level. We select the model checkpoint with the least perplexity on the Librispeech validation set.

**Fine-tuning Language Models**: LSTM and Transformer models were fine-tuned on the out-of-domain train set with a batch size of 256 for 25 epochs. We select the model checkpoint with the least perplexity on the out-of-domain validation set.

**ASR Audio Models**: For generating ASR n-best hypotheses, we use a pre-trained RNN-T model with the Emformer encoder (Shi et al., 2021),

| Dataset | Train | Val | Test | Vocab Size | Domain | # Domains |
|---|---|---|---|---|---|---|
| Earnings-21+22 | 49.6K | 7.1K | 14.2K | 20K | Earning Call | 169 |
| AMI Meeting Corpus | 17.1K | 2.7K | 5.8K | 11K | Meeting Recording | 135 |
| TED-LIUM v3 | 188.9K | 26.6K | 9.3K | 46K | TED Talk | 2351 |
| Wikitext-103 | 2M | 300K | 10K | 200K | Wikipedia Page | 30k |
| UserLibri | 6.3M | 700K | 10K | 10K | Books | 107 |

Table 7: Data stats of ASAP, UserLibri, and WikiText-103.

LSTM predictor, and a joiner with 80M parameters.

**PersonaLM Retrieval Augmentation**: Probability distribution obtained from bi-gram frequencies for each vocabulary word in the co-occurrence matrices was normalized. Trainable parameters in the Neural Retrieval module were frozen during inference and used to encode query and domain contexts. For each model, we report results with minimal perplexity by iterating the interpolation parameter $\lambda$ between 0 to 1 in increments of 0.1.

**Loss Functions**: `PersonaLM` retrieval augmentation uses both group-wise contrastive loss and MLM loss to train the neural retriever. For pretraining and fine-tuning the LMs, we use cross entropy loss over target words in the sequence.

**Model Parameters**: The audio LSTM model has around 80M parameters. Transformer LM has approx 247M parameters while the LSTM LM has around 6.5M parameters. The conformer model for streaming and non-streaming ASR in UserLibri dataset has 86M parameters.

**Evaluation Metrics**: We utilize word-level perplexity scores to evaluate language model performance for next-word prediction. We also report Word Error Rate (WER) for ASR second-pass rescoring in speech datasets.

**Data Preprocessing**: For each text sentence/utterance in the ASAP benchmark, all alphabets were lower-cased, numbers were converted into their respective word forms, and punctuation was removed.

**Pre-training SCAN Retreiver**: The maximum query and span lengths for the ranking datasets are set to 32 and 128, respectively. We experiment the learning rate between $\{1e - -5, 2e - 5, 3e - 5, 4e - 5, 5e - 5\}$, weight decay $\omega \in \{1e - 6, 1e - 5, 1e - 4, 1e - 3\}$, batch size $b \in \{4, 8, 16, 32, 64, 128, 256, 512\}$ and epochs ($\leq 25$). and Adam optimizer with a linear warm-up for the first 10% steps.

**Fine-tuning FlanT5-base for LaMP Experi-ment**: We utilize a FlanT5-base model for few-shot fine-tuning experiments with a learning rate experimented between $\{1e - -5, 2e - 5, 3e - 5, 4e - 5, 5e - 5\}$, weight decay $\omega \in \{1e - 6, 1e - 5, 1e - 4, 1e - 3\}$, batch size $b \in \{4, 8, 16, 32, 64, 128, 256, 512\}$ and epochs ($\leq 25$) and Adam optimizer. We utilize a linear warmup scheduler for 5% of the total training steps and train for a maximum of 20 epochs. The maximum input prompt length is 512 tokens.

## B.10 Implementation Software and Packages

We implemented our solution in Python 3.6 using the PyTorch framework. We used the following libraries and modules:

- Huggingface's implementation for BERT transformers. [10]

- FAISS Library [11] for fast kNN retrieval.

- Scipy [12] for scientific computations

- NLTK [13] for sentence processing

- OpenAI [14] for GPT-3.5 turbo

## B.11 Model Decisions

We choose optimal k in the kNN algorithm to be 400. We experiment with bigrams, trigrams, and 4-grams. The optimal weightage of the n-grams is $[(0.6 - 0.7), (0.10 - 0.20), (0.05 - 0.10)]$. We choose the maximum value of $T = 20$ from each document for training the SCAN retriever.

## B.12 LLM Prompts

We utilize the prompts given by (Salemi et al., 2023) in the paper Appendix for retrieval augmentation experiments. This was done to ensure a straightforward comparison with baseline BM25

---

[10]https://huggingface.co/
[11]https://github.com/facebookresearch/faiss
[12]https://scipy.org/
[13]https://www.nltk.org/
[14]https://platform.openai.com/docs/models/overview

and Contriever. The retrieved user profiles were added to the prompt context while ensuring the maximum context length is respected.

### B.13  Instructions given to participants or annotators

We do not annotate any new data.

### B.14  Annotator recruitment and payment

We do not annotate any new data.

### B.15  Consent for data annotation

We do not annotate any new data.

### B.16  Protocol approval by ethics review board

We do not annotate any new data.

### B.17  Demographic and geographic characteristics of the annotator population

We do not annotate any new data.