# OpenReview forum: "PersonaLM: Language Model Personalization via Domain-distributed Span Aggregated K-Nearest N-gram Retrieval Augmentation"
_EMNLP/2023/Conference — EMNLP 2023 Findings_

### Official Review · Reviewer_cwNA · 2023-08-04

**Soundness:** 3

**Excitement:**

3: Ambivalent: It has merits (e.g., it reports state-of-the-art results, the idea is nice), but there are key weaknesses (e.g., it describes incremental work), and it can significantly benefit from another round of revision. However, I won't object to accepting it if my co-reviewers champion it.

**Paper Topic And Main Contributions:**

The paper proposes an approach to improve language model next-word prediction performance (lower perplexities) for better ASR personalization. The main hypothesis (idea) is that n-gram patterns are domain/user-specific, and augmenting LM predictions with n-gram probabilities from a subset of query-relevant users/domains may lead to better personalization. The paper shows a significant drop in LM perplexities and improvement in ASR performance.

**Reasons To Accept:**

- One of the reasons would be the novelty of the idea and observed improvements.
- The paper is well written in the latter half.

**Reasons To Reject:**

I am merging reasons to reject and clarifications in one place:

1) L061-064: "However, these methods give subpar performance as they do not retrieve relevant domains/users prior to context selection from billions of candidates."-I am not sure if kNN-LM and follow-up works would even have that issue as they are not meant to solve the user personalization task. kNN-LM was proposed primarily to enhance the memorization capabilities of LM to generalize better i.e. lower perplexity. So I would suggest rephrasing this line to enhance readability. The problem is clear but the way it is motivated does not look very appropriate (by reading paragraph L058-068).

2) What is K in the equation in line 164? Are you marginalizing retrieved documents? I would prefer authors refer to the RAG paper to make the mathematical details more sound and intuitive.

3) Until after section 3, I am still unclear about the definition of domain, I think the notations d, D, Domain_1, are confusing, leading to a confusing methodology. I could not find the definition of the query as well, where do you obtain w1...w(t-1) from?

4) Do you use positional encoding in SCAN retriever (encoder) input? If no---then in figure-3 Domain_1, Domain_2 positions would not matter much (which is good) however, how do you feed word position information? If yes---wouldn't the Domain_i position will impact its representation?

I think the paper can be significantly improved in the explanation of methodology and motivation. Definitions of some basic yet crucial terms are missing. Hoping to get more clarification on the methodology in the author's response.


**Reproducibility:**

3: Could reproduce the results with some difficulty. The settings of parameters are underspecified or subjectively determined; the training/evaluation data are not widely available.

**Reviewer Confidence:**

3: Pretty sure, but there's a chance I missed something. Although I have a good feel for this area in general, I did not carefully check the paper's details, e.g., the math, experimental design, or novelty.

---

> ### Author Rebuttal · Authors · 2023-08-28
>
> We thank the reviewer for his thoughtful comments and suggestions.
>
> ***Q-1. Use of kNN-LM for LM personalization:*** kNN-LM can retrieve and augment relevant context with LM predictions, thus potentially useful for LM personalization. We utilized kNN-LM and Neural Cache (see Sec-4.3 Baselines) due to their capability to memorize context which can help lower perplexity for user/domain-specific text, thereby improving ASR word error rate. However, we agree that kNN-LM is not a direct application for user personalization tasks. We would incorporate the reviewer's suggestion to rephrase the motivation paragraph to enhance readability.
>
> ***Q-2. Typo in line 164:*** K in the equation on line 164 refers to “K” domains. Yes, we are marginalizing retrieved documents. We will fix the typo in Line 163 - “domains ($d_1, d_2, · · ·, d_N $)” to “domains ($d_1, d_2, · · ·, d_K$)”.  We thank the author for suggestions to improve the mathematical intuition of LM conditional probability based on the marginalization of retrieved domains. We will refer to the RAG paper [1] to make the mathematical details more sound and intuitive in the camera-ready version of the paper.
>
> [1] Lewis, Patrick, et al. "Retrieval-augmented generation for knowledge-intensive nlp tasks." Advances in Neural Information Processing Systems 33 (2020): 9459-9474.
>
> ***Q-3.Definition of domain:***  Domains in our work refer to the categories that have a distribution different from the base data used to train the base model. It can mean different users/topics/call recordings based on the dataset. Domains in the AMI Meeting corpus are formed based on speaker IDs. Domains in Earnings-21+22 data correspond to different companies. A specific domain in the TED-LIUM v3 dataset refers to a particular user. Domains in Wikitext-103 correspond to individual Wikipedia pages. We will clarify all these details in Table 1 as well as Section 4.2 to make it easy for the readers. Domains are represented by a small letter “$d$” in our methodology. “$d^+$” and “$d^-$” refer to matching and contrasting domains with respect to the domain of the query. We will ensure that all mentions of domains have a consistent notation throughout the paper in the final version. Line 155 defines query as a sequence of words “$q = (w_1, · · ·, w_{t−1})$” for which the next word “$w_t$” is to be predicted.
>
> ***Q-4. Position encoding in SCAN retriever:*** We utilize BERT to initialize the SCAN retriever as mentioned in Sec-4.1. Hence the SCAN architecture uses a learnable position embedding. We use [SEP] tokens to identify the representation of the spans in the input sequence. While training the SCAN retriever, we train the Transformer model with permutations of span order. Hence, the spans from positive domains are not necessary to always precede the spans from negative domains. Similarly, the spans from an individual domain are not necessitated to be always together in the sequence. This ensures the learned span representations are robust in terms of their relative order. We will clarify this more in Sec-4.1.

---

### Official Review · Reviewer_FqGg · 2023-08-05

**Soundness:** 4

**Excitement:**

3: Ambivalent: It has merits (e.g., it reports state-of-the-art results, the idea is nice), but there are key weaknesses (e.g., it describes incremental work), and it can significantly benefit from another round of revision. However, I won't object to accepting it if my co-reviewers champion it.

**Paper Topic And Main Contributions:**

The paper proposes a retrieval-based approach to recharacterize the next-word probability in LMs as the probability of which domain the query is in and the vocabulary probability conditioned on the relevance of the underlying domains. For the former, they design a neural retriever to calculate embeddings for the query and each span of domains, while the latter, they  design a n-gram co-occurrence matrix to learn the word frequency within domains. The experiments show that the perplexity and word error rate are both improved with the proposed method.

**Questions For The Authors:**

1. For SCAN retriever, it seems to be equivalent to extract embeddings of random texts for each domains and contrast them with the current query embedding. Would it be more straightforward to directly encode each sentence with a shared encoder and apply a contrastive loss similar in SimCSE?
2. Could you explain again how you calculate the N-gram co-occurence matrix? Do you mean for each vocabulary word, you query top-k n-grams in a specific domain based on the vicinity of Bert embeddings within n-grams?
3. What are the domains that are adopted for k-NN based retrieval in each dataset?

**Reasons To Accept:**

1. The paper proposes a new group-contrastive neural retriever for ranking query-relevant external domains which are beneficial to improve LM prediction.
2. The experiments are extensive to both show how the performance of the proposed framework is compared with other competitive baselines and how modified next-word probability distribution may affect the generation quality.

**Reasons To Reject:**

1. Some experimental details should be further clarified to support the claim of the proposed method.
2. The narration could be better improved where some sections are a little bit hard to follow such as Section 3.3.

**Reproducibility:**

4: Could mostly reproduce the results, but there may be some variation because of sample variance or minor variations in their interpretation of the protocol or method.

**Reviewer Confidence:**

3: Pretty sure, but there's a chance I missed something. Although I have a good feel for this area in general, I did not carefully check the paper's details, e.g., the math, experimental design, or novelty.

---

> ### Author Rebuttal · Authors · 2023-08-28
>
> We thank the reviewer for their thoughtful feedback. We clarify the questions as follows:
>
> ***Q-1 SCAN Retriever Embedding***: The SCAN retriever is a significant improvement over the Dense Passage Retriever style shared encoder learned through SimCSE in terms of its training methodology. The core novelties that differentiate it from SimCSE-based contrastive retriever are as follows:
>
> 1. SCAN retriever is trained to contrast between spans from different domains by encoding the spans in the Transformer input layer itself (see Sec-3.1 Line 205-210). As a result, it can take maximum advantage of the Transformer’s self-attention mechanism to discriminate between spans from unrelated domains.
>
> 2. The group contrastive loss requires fewer iteration steps to converge as the model can encode multiple spans from both positive and negative domains (See Sec-3.1 Line 220-225) as opposed to SimCSE loss that relies on a single positive span from the batch of negative spans.
>
> 3. The multi-granular encoding strategy helps SCAN to learn hierarchical concepts related to specific words, entities, phrases, and topics information (See Sec-3.1 Line 180-187). Shared encoders with SimCSE loss rely on averaging Bert vectors of the entire domain during training which weakens the representation power of the retriever.
>
> Quantitative results (See Tables 1-3) show how using DANK-LM with Dense Passage Retriever leads to suboptimal performance across different datasets. SCAN retriever is ~ 5% better than DPR across different settings.
>
> ***Q-2 Construction of N-gram Frequency Matrix***: The N-gram frequency matric is a matrix where a cell (i,j) corresponds to the normalized frequency of observing word w_i ∈ V followed by a sequence of n-1 words {${w_j-n, w_j-n+1,..., w_j}$} in a specific domain. N-gram frequency matrices can be sparse and constructing large matrices is memory-inefficient (as mentioned in Sec-3.3 Line 252). Restricting the query n-grams to only exact matches leads to a loss of information due to ignoring semantically similar n-grams. Hence, we build domain-specific n-gram datastores that store key-value pairs of ((n-gram Bert vector, next word), co-occurrence frequency)). For each vocabulary word, we query top-k n-grams in a specific domain based on the vicinity of Bert embeddings of the n-gram within the datastore to augment with LM predictions. We will make this description clearer in Sec-3.3 of the camera-ready version to enhance the understanding of the readers.
>
> ***Q-3 Domains in KNN-based Retrieval***: Our method is extensible to different types of domains (users/topics/categories) as highlighted in Table 1 and Sec-4.2. Domains in the AMI Meeting corpus are formed based on speaker IDs. Domains in Earnings-21+22 data correspond to different companies. A specific domain in the TED-LIUM v3 dataset refers to a particular TED talk speaker. Domains in Wikitext-103 correspond to individual Wikipedia pages.

---

### Official Review · Reviewer_n2GV · 2023-08-11

**Soundness:** 3

**Excitement:**

3: Ambivalent: It has merits (e.g., it reports state-of-the-art results, the idea is nice), but there are key weaknesses (e.g., it describes incremental work), and it can significantly benefit from another round of revision. However, I won't object to accepting it if my co-reviewers champion it.

**Paper Topic And Main Contributions:**

This work proposed a contextual biasing method based on domain specific n-grams, which are interpolated with a neural LM. In this method, a query span is required to obtain domain importance weights. This method is applicable to asr rescoring.

**Questions For The Authors:**

It's not clear from the paper what is defined as a domain for each of the sets in table 1. Is it simply different speakers? The use of speaker/domain throughout the paper is a bit confusing.

**Reasons To Accept:**

The method appears novel. The results are convincing on multiple datasets.

**Reasons To Reject:**

Section 4.2 is not easy to understand.

**Reproducibility:**

2: Would be hard pressed to reproduce the results. The contribution depends on data that are simply not available outside the author's institution or consortium; not enough details are provided.

**Reviewer Confidence:**

1: Not my area, or paper was hard for me to understand. My evaluation is just an educated guess.

**Typos Grammar Style And Presentation Improvements:**

"Improving neural language models with a continuous cache." citation is missing a year.

---

> ### Author Rebuttal · Authors · 2023-08-28
>
> We thank the reviewer for their thoughtful feedback and constructive suggestions.
>
> ***Domains for each set in Table-1***: Our proposed method - DANK-LM is applicable to a variety of different datasets with varying “domains” meaning different speakers/topics/categories. For example, the recordings in the AMI Meeting corpus are split based on speaker IDs. Domains in Earnings-21+22 data correspond to different companies. A specific domain in the TED-LIUM v3 dataset refers to a TED talk speaker. Domains in Wikitext-103 correspond to individual Wikipedia pages. We will clarify these details in Table 1 and add them in Section 4.2 to make it easy for the readers. We will also release the data splits for enhanced reproducibility.
>
> ***Improvements in Sec-4.2***: We will expand Section-4.2 to clarify the data split, domain selection, and any domain-specific preprocessing steps by utilizing the additional page provided to authors in the camera-ready version.
>
> ***Missing Year in Citation Typo***: We thank the reviewer for pointing out the typo mistake on line 375 - "Improving neural language models with a continuous cache." The correct citation is “Edouard Grave, Armand Joulin, and Nicolas Usunier. Improving neural language models with a continuous cache. In 5th International Conference on Learning 740 Representations, ICLR 2017, Toulon, France”. We will add the missing year in the citation in the final version of the paper.

---

### Meta-Review · Area_Chair_MV4V · 2023-09-19

**Recommendation:** 3

**Metareview:**

**Originality:**

This paper describes a novel method for domain / user personalization via language model adaptation using a domain encoder trained with a contrastive loss.

**Signifciance:**

Personalizing language models to domains/users, especially in order to accurately recognize proper nouns and other domain specific vocabulary is very important. This work describes a method that achieves impressive performance improvement by conditioning on cached n-gram statistics from detected close domains.

**Clarity:**

The consensus appears to be that the paper is somewhat difficult to follow. In general the explanation of methodology and motivation could be improved, including definitions of some basic yet crucial terms which are missing. Furthermore, Section 4.2 difficult to understand given the diverse datasets and task construction described, and much of Section 3.3, which one reviewer found difficult to follow, might have benefited from description in algorithmic, or mathematical terms rather than a purely textual description.

**Pros**
   - Addresses a relevant and interesting problem
   - Novel model
   - Very extensive experiments
   - Thoroughly described training and decoding parameters and experimental setup
   - Impressive results on multiple domains

**Cons**
   - A very complicated experimental setup which may be difficult to reproduce, in spite of the fact that it is well described.
   - For instance the ASAP task is well defined, but to reproduce results would require recreating the dataset setup from scratch as it is not released as a part of the paper,  nor is the code-base to reproduce results
   - The paper, as currently written, could significantly improve the narration, motivation and explanation of the data and experimental setup

---

### Decision · Program_Chairs · 2023-10-07

**Decision:**

Accept-Findings

**Comment:**

**Originality:**

This paper describes a novel method for domain / user personalization via language model adaptation using a domain encoder trained with a contrastive loss.

**Signifciance:**

Personalizing language models to domains/users, especially in order to accurately recognize proper nouns and other domain specific vocabulary is very important. This work describes a method that achieves impressive performance improvement by conditioning on cached n-gram statistics from detected close domains.

**Clarity:**

The consensus appears to be that the paper is somewhat difficult to follow. In general the explanation of methodology and motivation could be improved, including definitions of some basic yet crucial terms which are missing. Furthermore, Section 4.2 difficult to understand given the diverse datasets and task construction described, and much of Section 3.3, which one reviewer found difficult to follow, might have benefited from description in algorithmic, or mathematical terms rather than a purely textual description.

**Pros**
   - Addresses a relevant and interesting problem
   - Novel model
   - Very extensive experiments
   - Thoroughly described training and decoding parameters and experimental setup
   - Impressive results on multiple domains

**Cons**
   - A very complicated experimental setup which may be difficult to reproduce, in spite of the fact that it is well described.
   - For instance the ASAP task is well defined, but to reproduce results would require recreating the dataset setup from scratch as it is not released as a part of the paper,  nor is the code-base to reproduce results
   - The paper, as currently written, could significantly improve the narration, motivation and explanation of the data and experimental setup